# Interpreting blood GLUcose data with R package iglu

**Steven Broll[1¤a], Jacek Urbanek[2], David Buchanan[1], Elizabeth Chun[3], John Muschelli[4], Naresh M. Punjabi[2¤b], Irina Gaynanova[1]** *

**1** Department of Statistics, Texas A&M University, College Station, TX, United States of America, **2** School of Medicine, Johns Hopkins University, Baltimore, MD, United States of America, **3** Department of Biology, Texas A&M University, College Station, TX, United States of America, **4** Johns Hopkins Bloomberg School of Public Health, Johns Hopkins University, Baltimore, MD, United States of America

¤a Current address: Department of Statistics and Data Science, Cornell University, Ithaca, NY, United States of America
¤b Current address: Miller School of Medicine, University of Miami, Miami, FL, United States of America
* irinag@stat.tamu.edu

**Data Availability Statement:** The data underlying the results presented in the study are available as part of the iglu package. The package can be accessed from https://CRAN.R-project.org/package=iglu.

## Abstract

Continuous Glucose Monitoring (CGM) data play an increasing role in clinical practice as they provide detailed quantification of blood glucose levels during the entire 24-hour period. The R package `iglu` implements a wide range of CGM-derived metrics for measuring glucose control and glucose variability. The package also allows one to visualize CGM data using time-series and lasagna plots. A distinct advantage of `iglu` is that it comes with a point-and-click graphical user interface (GUI) which makes the package widely accessible to users regardless of their programming experience. Thus, the open-source and easy to use `iglu` package will help advance CGM research and CGM data analyses. R package `iglu` is publicly available on CRAN and at https://github.com/irinagain/iglu.

## Introduction

Continuous Glucose Monitors (CGMs) are small wearable devices that record measurements of blood glucose levels at frequent time intervals. As CGM data provide a detailed quantification of the variation in blood glucose levels, CGMs play an increasing role in clinical practice [1]. While multiple CGM-derived metrics to assess the quality of glycemic control and glycemic variability have been developed [2], their complexity and variety pose computational challenges for clinicians and researchers. While some metrics (e.g. mean) can be directly calculated from the data, others require additional pre-processing steps, such as projecting glucose measurements on an equidistant time grid (e.g. CONGA, SDbdm) or the imputation of missing data.

We are aware of two existing R packages for CGM data analyses: CGManalyzer [3] and cgmanalysis [4]. These packages are primarily designed to read and organize CGM data, rather than provide an easy-to-use interface for a comprehensive evaluation of available CGM characteristics. While their analytical utility is undeniable, a substantial number of CGM metrics summarized in [2] is not available. Moreover, both packages require the users to have

**Funding:** This work was supported by the National Institutes of Health [HL11716, HL146709 to N.P.]; and an agreement from the Johns Hopkins University [80045538 to I.G.]. The funders had no role in study design, data collection and analysis, decision to publish, or preparation of the manuscript.

**Competing interests:** The authors have declared that no competing interests exist.

considerable programming experience, which might be a limiting factor for researchers seeking robust and accessible analytical solutions. `EasyGV` is another free CGM software in the form of a macro-enabled Excel workbook [5], and thus is more accessible compared to `CGManalyzer` and `cgmanalysis`. However, it only allows calculation of 10 metrics. Furthermore, unlike R, Excel is not a script-based programming language, which makes it less desirable for those users who want to create reproducible scripts for all data processing and metric calculation steps. Thus, there remains a need for open-source software that (i) computes most of the CGM metrics available from the literature, and (ii) meets the needs of researchers with varying levels of programming experience.

Our R package `iglu` calculates all CGM metrics summarized in [2] in addition to several others [6, 12, 19], a full list of currently implemented metrics is summarized in Table 1. A comparison of functionality with `CGManalyzer` [3] and `cgmanalysis` [4] is in Table 2. Additional improvements include advanced visualization with lasagna plots [21], and provided example CGM datasets that make it easy to get started. Finally, a distinct advantage of `iglu` over existing open-source CGM software is a point-and-click graphical user interface (GUI) which makes the package accessible to users with little to no R experience.

## Features

### Example data

The `iglu` package is designed to work with CGM data provided in the form of a data frame with three columns: id (subject identifier), time (date and time stamp) and gl (corresponding blood glucose measurement in mg/dL). The package comes with two example datasets that follow this structure. `example_data_5_subject` contains Dexcom G4 CGM measurements from subjects with Type II diabetes.

```
example_data_5_subject[1:2,]
     id                time     gl
1 Subject 1 2015-06-06 16:50:27 153
2 Subject 1 2015-06-06 17:05:27 137
```

`example_data_1_subject` is a subset corresponding to one subject. These data are part of a larger study analyzed in [22].

### Illustration of metrics use

Table 1 summarizes all the metrics implemented in the package, which can be divided into two categories: time-independent and time-dependent. All the functions assume that the glucose value are given in mg/dL units. Each function has detailed documentation that describes all the input parameters (and their default values) as well as the specific algorithm used for metric calculation. Full documentation can be accessed from the R console after loading the `iglu` package (e.g. `? active_percent`) or from the accompanying website (https://irinagain.github.io/iglu/).

One example of a time-independent metric is Hyperglycemia index [2], the corresponding `iglu` function returns a single value for each subject in a tibble object [23]. Subject id will always be printed in the id column, and metrics will be printed in the following columns.

```
hyper_index(example_data_5_subject)
# A tibble: 5 x 2
  id          hyper_index
  <fct>           <dbl>
1 Subject 1       0.391
2 Subject 2       4.17
```

**Table 1. Summary of CGM metrics implemented in iglu.**

| Metric name | iglu function | Reference | Time-dependent |
|---|---|---|---|
| Active percent | active_percent | Danne et al. (2017) [6] | Yes |
| ADRR | adrr | Kovatchev et al. (2006) [7] | Yes |
| AUC | auc | Danne et al. (2017) [6] | Yes |
| COGI | cogi | Leelarathna et al. (2019) [8] | No |
| CONGA | conga | McDonnell et al. (2005) [9] | Yes |
| CV | cv_glu | Rodbard (2009) [2] | No |
| CV subtypes | cv_measures | Umpierrez & Kovatchev (2018) [10] | Yes |
| eA1c | ea1c | Nathan et al. (2008) [11] | No |
| GMI | gmi | Bergenstal et al. (2018) [12] | No |
| GRADE | grade | Hill et al. (2007) [13] | No |
| GRADEeu | grade_eugly | Hill et al. (2007) [13] | No |
| GRADEhyper | grade_hyper | Hill et al. (2007) [13] | No |
| GRADEhypo | grade_hypo | Hill et al. (2007) [13] | No |
| GVP | gvp | Peyser et al. (2018) [14] | Yes |
| HBGI | hbgi | Kovatchev et al. (2006) [7] | No |
| LBGI | lbgi | Kovatchev et al. (2006) [7] | No |
| Hyper Index | hyper_index | Rodbard (2009) [2] | No |
| Hypo Index | hypo_index | Rodbard (2009) [2] | No |
| IGC | igc | Rodbard (2009) [2] | No |
| IQR | iqr_glu | | No |
| J-index | j_index | Wojcicki (1995) [15] | No |
| MAD | mad_glu | | No |
| MAG | mag | Hermanides et al. (2010) [16] | Yes |
| MAGE | mage | Service & Nelson (1980) [17] | No |
| Mean | mean_glu | | No |
| Median | median_glu | | No |
| MODD | modd | Service & Nelson (1980) [17] | Yes |
| M-value | m_value | Schlichtkrull et al. (1965) [18] | No |
| Percent Above | above_percent | | No |
| Percent Below | below_percent | | No |
| Percent in range | in_range_percent | | No |
| Quantiles | quantile_glu | | No |
| Range | range_glu | | No |
| ROC (Rate Of Change) | roc | Clarke & Kovatchev (2009) [19] | Yes |
| SD of ROC | sd_roc | Clarke & Kovatchev (2009) [19] | Yes |
| SD | sd_glu | | No |
| SD subtypes | sd_measures | Rodbard (2009) [20] | Yes |

```
3 Subject 3       1.18
4 Subject 4       0.358
5 Subject 5       2.21
```

In this example, Subject 2 has the largest Hyperglycemia index, indicating the worst hyperglycemia. This is reflected in percent of times Subject 2 spends above fixed glucose targets.

```
above_percent(example_data_5_subject)
# A tibble: 5 x 4
  id        above_140  above_180 above_250
  <fct>         <dbl>      <dbl>     <dbl>
```

**Table 2. Comparison of iglu functionality with existing R packages for CGM data.**

| Metric name | CGManalyzer | cgmanalysis | iglu |
|---|:---:|:---:|:---:|
| Active percent | × | ✓ | ✓ |
| ADRR | × | × | ✓ |
| AUC | × | ✓ | ✓ |
| COGI | × | × | ✓ |
| CONGA | ✓ | ✓ | ✓ |
| CV subtypes (mean, sd) | × | ✓ | ✓ |
| CV median | × | ✓ | × |
| eA1c | × | ✓ | ✓ |
| Excursions count (over/under) | × | ✓ | × |
| GMI | × | ✓ | ✓ |
| GRADE | × | × | ✓ |
| GVP | × | × | ✓ |
| HBGI/LBGI | × | ✓ | ✓ |
| IGC | × | × | ✓ |
| J-index | × | ✓ | ✓ |
| MAG | × | × | ✓ |
| MAGE | × | ✓ | ✓ |
| MODD | ✓ | ✓ | ✓ |
| M-value | × | × | ✓ |
| Multiscale entropy | ✓ | × | × |
| Percent in range | × | ✓ | ✓ |
| Time in range | ✓ | ✓ | × |
| ROC (Rate of Change) | × | × | ✓ |
| SD subtypes | × | × | ✓ |
| Summary statistics | ✓ | ✓ | ✓ |
| Day/night metrics (SD, min, max, AUC) | × | ✓ | × |
| GUI for calculation | × | × | ✓ |

```
1 Subject 1    26.7     8.40    0.446
2 Subject 2    96.8    74.4    26.7
3 Subject 3    51.5    18.9     5.74
4 Subject 4    32.9     4.97    0
5 Subject 5    70.8    38.1    11.6
```

The default target values of 140, 180 and 250 mg/dL in `above_percent` can be adjusted by the user.

Examples of time-dependent metrics include measures of glycemic variability such as CONGA [9] and standard deviation of rate of change [19]. In the example data, the standard deviation of rate of change is the highest for Subject 5:

```
sd_roc(example_data_5_subject)
# A tibble: 5 x 2
  id        sd_roc
  <fct>     <dbl>
1 Subject 1 0.620
2 Subject 2 0.642
3 Subject 3 0.831
4 Subject 4 0.617
```

```
5 Subject 5 1.05
```

This provides an additional level of CGM data interpretation, since frequent or large glucose fluctuations may contribute to diabetes-related complications independently from chronic hyperglycemia [24]. Other metrics of glycemic variability confirm the high fluctuations in Subject 5, with all but one of the subtypes of standard deviation being the largest for Subject 5 [20]:

```
sd_measures(example_data_5_subject)
# A tibble: 1 x 7
  id          SdW SdHHMM SdWSH SdDM   SdB SdBDM
  <fct>      <dbl>  <dbl> <dbl> <dbl> <dbl> <dbl>
1 Subject 1 26.4   19.6  6.54  16.7  27.9  24.0
2 Subject 2 36.7   22.8  7.62  52.0  48.0  35.9
3 Subject 3 42.9   14.4  9.51  12.4  42.8  42.5
4 Subject 4 24.5   12.9  6.72  16.9  25.5  22.0
5 Subject 5 50.0   29.6  12.8  23.3  50.3  45.9
```

The calculations of these variability metrics require evenly spaced glucose measurements across time; however, this is not always the case in practice due to missing values and misalignment of CGM measurement times across subjects (e.g. measurement at 17:30 for Subject 1, but at 17:31 for Subject 2). In order to create a uniform evenly spaced grid of glucose measurements, iglu provides the function CGMS2DayByDay. This function is automatically called for metrics requiring the evenly spaced grid across days, however the user can also access the function directly. The function works on a single subject's data, and has three outputs.

```
str(CGMS2DayByDay(example_data_1_subject))
List of 3
  $ gd2d        : num [1:14, 1:288] NA 112.2 92 90.1 143.1 ...
  $ actual_dates: Date[1:14], format: "2015-06-06"
"2015-06-07" ...
  $ dt0         : num 5
```

The first part of the output, gd2d, is the interpolated grid of values. Each row corresponds to one day of measurements, and the columns correspond to an equi-distant time grid covering a 24 hour time span. The grid is chosen to match the frequency of the sensor (5 minutes in this example leading to (24*60)/5 = 288 columns), which is returned as dt0. The linear interpolation is only performed between observed CGM values that are less than inter_gap minutes apart, otherwise missing values are inserted. By default, the function uses inter_gap = 45 minutes, however this value can be adjusted by the user. The returned actual_dates allows one to map the rows in gd2d back to original dates. The achieved alignment of glucose measurement times across the days enables both the calculation of corresponding metrics, and the creation of lasagna plots discussed in the next section.

Finally, iglu also allows one to assess the reliability of estimated CGM metrics by providing information on the number of days of data collection together with % of time the CGM device was active during those days (% of non-missing measurements). This information is automatically provided as part of the standardized AGP output discussed in the next section, and can also be obtained directly by calling the function active_percent.

```
active_percent(example_data_5_subject)
# A tibble: 5 x 5
  id        active_percent ndays  start_date           end_date
  <fct>          <dbl> <drtn>      <dttm>               <dttm>
1 Subject 1   79.8 12.7 days  2015-06-06 16:50:27 2015-06-19
08:59:36
```

```
  2 Subject 2    58.9 16.7 days   2015-02-24 17:31:29 2015-03-13
09:38:01
  3 Subject 3    92.1  5.8 days   2015-03-10 15:36:26 2015-03-16
10:11:05
  4 Subject 4           98.7 12.9 days 2015-03-13 12:44:09 2015-
03-26 10:01:58
  5 Subject 5           95.8 10.6 days 2015-02-28 17:40:06 2015-
03-11 08:04:28
```

According to [25], 10-14 days of CGM measurements are generally sufficient for assessing outcomes in clinical trials, and for determining potential adjustments to diabetes management based on retrospective review. Given these recommendations, the estimates of CGM parameters for Subject 3 are less reliable than the estimates for other subjects in the example dataset.

To investigate the agreement of metrics calculations with existing software, we selected a subset of metrics for cross-comparison of CGManalyzer, cgmanalysis and iglu on the example dataset. We found that the summary statistics (min, max, mean, quantiles, total SD) are in perfect agreement for all 5 subjects across all three packages. Additionally, cgmanalysis and iglu have perfect agreement in the values of GMI, eA1C, CV, % of glucose values in range and J-index. There is a slight (less than 1%) disagreement in % of time CGM is active between iglu and cgmanalysis, which we suspect is due to varying rounding precision. There is also some disagreement in all three packages in CONGA values (using a common parameter of $n = 1$ hour), which we suspect is due to differences in handling missing values and in grid interpolation schemes. Overall, the results show good agreement between the three packages as the calculated metrics either match perfectly or are close. Table 3 shows explicit values for Mean, SD, % time CGM is active, GMI, J-index and CONGA ($n = 1$ hour) for all five subjects across all three packages.

## Visualizations

The iglu package has several visualization capabilities, which are summarized in Table 4. The main function is plot_glu, which by default provides a time series plot for each subject. The glucose values are plotted on a linear scale, however an optional log parameter can be used to display glucose on a semilogarithmic scale [26]. Fig 1 illustrates the default output on example data with the horizontal red lines indicating user-specified target range, the default range is [70, 180] mg/dL [27]. The visual inspection of the plots confirm the previous conclusions from comparison of Hyperglycemia index and metrics of glycemic variability across subjects: the majority of measurements for Subject 2 are above 180 mg/dL, however the variability is larger for Subject 5.

Another visualization type is provided via lasagna plots [21], which use a color grid rather than a number scale to visualize trends in data over time. The lasagna plots in iglu can be single-subject or multi-subject. The single-subject lasagna plot has rows corresponding to each day of measurements with a color grid indicating glucose values (Fig 2A). An optional within-time sorting across days allows one to investigate average glucose patterns as a function of 24 hour time periods (Fig 2B). The multi-subject lasagna plot has rows corresponding to subjects, with a color grid indicating glucose values across the whole time domain, or average glucose values across days. The highest glucose values are displayed in red, whereas the lowest are displayed in blue. Thus, the numerical glucose values are mapped to color using the gradient from blue to red (Fig 2), which corresponds to the default 'blue-red' color scheme. An alternative 'red-orange' color scheme can be selected by the user by corresponding modification of the 'color_scheme' parameter (using the gradient from red to green to yellow

**Table 3. Comparison of selected metrics across R packages using example dataset.**

| Metric name | Subject id | CGManalyzer | cgmanalysis | iglu |
|---|---|---|---|---|
| Mean | Subject 1 | 123.7 | 123.7 | 123.7 |
| | Subject 2 | 218.5 | 218.5 | 218.5 |
| | Subject 3 | 154.0 | 154.0 | 154.0 |
| | Subject 4 | 129.7 | 129.7 | 129.7 |
| | Subject 5 | 174.6 | 174.6 | 174.6 |
| SD | Subject 1 | 33.3 | 33.3 | 33.3 |
| | Subject 2 | 52.4 | 52.4 | 52.4 |
| | Subject 3 | 44.8 | 44.8 | 44.8 |
| | Subject 4 | 29.1 | 29.1 | 29.1 |
| | Subject 5 | 55.6 | 55.6 | 55.6 |
| % Time CGM is Active | Subject 1 | × | 79.0 | 79.8 |
| | Subject 2 | × | 58.0 | 58.9 |
| | Subject 3 | × | 92.0 | 92.1 |
| | Subject 4 | × | 98.0 | 98.7 |
| | Subject 5 | × | 95.0 | 95.8 |
| GMI | Subject 1 | × | 6.3 | 6.3 |
| | Subject 2 | × | 8.5 | 8.5 |
| | Subject 3 | × | 7.0 | 7.0 |
| | Subject 4 | × | 6.4 | 6.4 |
| | Subject 5 | × | 7.5 | 7.5 |
| J-index | Subject 1 | × | 24.6 | 24.6 |
| | Subject 2 | × | 73.3 | 73.3 |
| | Subject 3 | × | 39.5 | 39.5 |
| | Subject 4 | × | 25.2 | 25.2 |
| | Subject 5 | × | 54.4 | 54.4 |
| CONGA ($n = 1$ hour) | Subject 1 | 24.7 | 25.7 | 25.9 |
| | Subject 2 | 19.9 | 25.1 | 25.7 |
| | Subject 3 | 38.2 | 41.0 | 39.5 |
| | Subject 4 | 23.2 | 22.6 | 23.3 |
| | Subject 5 | 49.0 | 50.0 | 49.3 |

to orange, with green corresponding to values in specified glucose range). Fig 2C displays a customized multi-subject lasagna plot for example data that displays average glucose values across days for each subject; this plot is produced by the following call.

```
plot_lasagna(example_data_5_subject, datatype = "average",
                           midpoint = 140, limits = c(60,
400))
```

**Table 4. Summary of iglu visualization capabilities.**

| Function call | Visualization description | Main parameters |
|---|---|---|
| plot_glu | Multiple plot types: time series and lasagna | plottype, lasagnatype |
| plot_lasagna | Lasagna plot of glucose values for multiple subjects | datatype, lasagnatype |
| plot_lasagna_1subject | Lasagna plot of glucose values for a single subject | lasagnatype |
| plot_roc | Time series of glucose values colored by rate of change (ROC) | subjects, timelag |
| hist_roc | Histogram of rate of change (ROC) values | subjects, timelag |
| agp | Ambulatory Glucose Profile (AGP) | maxd, daily |

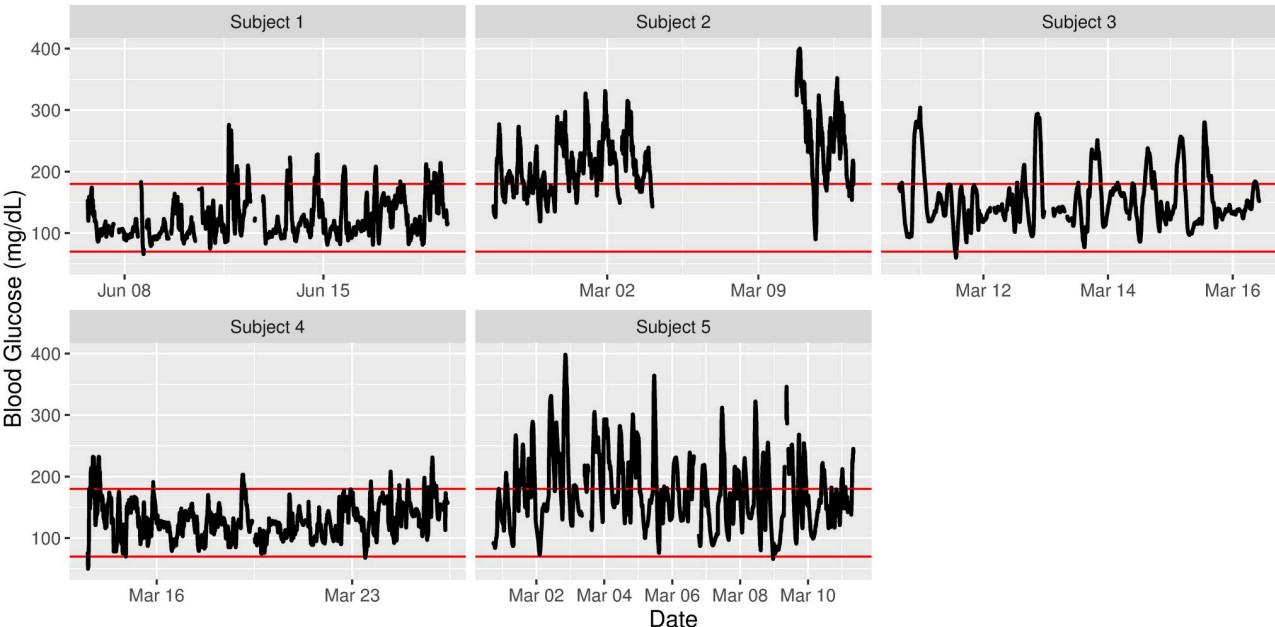

**Fig 1. Time series plots for five subjects.** Selected target range is [70, 180] mg/dL.

The `midpoint` specifies the glucose value (in mg/dL) at which the color transitions from blue to red (the default is 105 mg/dL), whereas the `limits` specify the range (the default is [50, 500] mg/dL). From Fig 2 one can for example infer that the glucose values for Subject 1 tend to be the highest in late afternoon (≈ 15:00—20:00). One can also infer that Subject 1

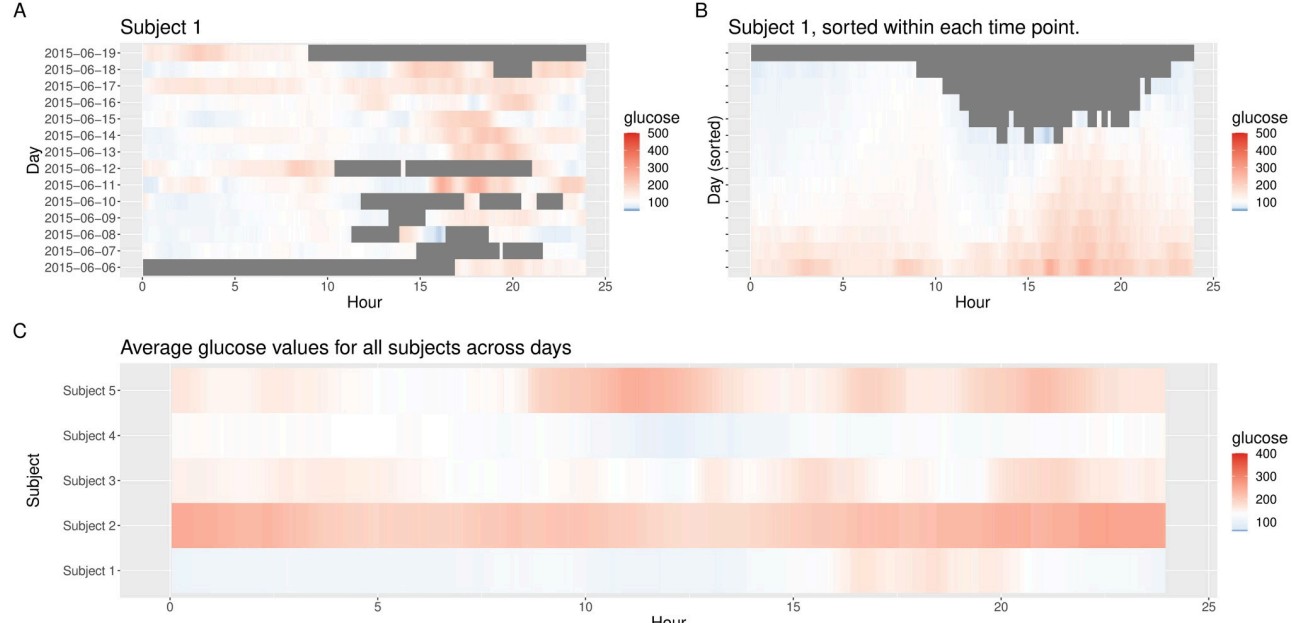

**Fig 2. Lasagna plots. (A)** unsorted and **(B)** time-sorted lasagna plot for Subject 1; **(C)** unsorted customized multi-subject lasagna plot based on average values across days.

tends to have the lowest glucose values during night time hours (0:00—6:00) compared to the other four subjects.

While lasagna plots are very similar to stacked bar charts introduced in [28], there are two main differences. First, the stacked bar charts split the glucose values into a fixed number of categories (based on specified glucose cutoffs), where the same color is used within each category. In contrast, the lasagna plots use gradient fill, thus the gradient of the color changes continuously with the change in glucose values. We believe this provides more detailed information on the subject's glucose profile. Secondly, the stacked bar charts in [28] are created for one subject at a time. In contrast, `iglu` allows one to create lasagna plots for multiple subjects at once. Using `datatype = 'average'` with `lasagnatype = 'subject-sorted'` facilitates direct cross-comparison of glucose distributions across subjects, whereas `lasagnatype = 'timesorted'` facilitates assessment of population-level trends. Fig 3 shows both types of plots. Fig 3A shows that Subject 2 has the highest levels of hyperglycemia, whereas Subjects 1 and 4 have the lowest levels of hyperglycemia. Fig 3B shows that among the 5 subjects, hyperglycemia is most common in the later afternoon, with the times around 4pm (16:00) and 9pm (21:00) showing the highest glucose values.

In addition to visualizing absolute glucose values, `iglu` also allows one to visualize local changes in glucose variability as measured by rate of change [19]. There are two types of visualizations associated with rate of change. The first is a time series plot of glucose values where each point is colored by the rate of change at that given time. Points colored in white have a stable rate of change, meaning the glucose is neither significantly increasing nor decreasing at that time point. Points colored red or blue represent times at which the glucose is significantly rising or falling, respectively. Thus colored points represent times of glucose variability, while white points represent glucose stability. Fig 4A shows a side by side comparison of rate of change time-series plots for two subjects. Subject 1 shows significantly less glucose variability than Subject 5. The function call to produce this plot is as follows.

```
plot_roc(example_data_5_subject, subjects = c("Subject 1",
"Subject 5"))
```

Fig 4B shows a side by side comparison of rate of change histogram plots for the same subjects. Once again, the colors show in what direction and how quickly the glucose is changing. The histogram plots allow one to immediately assess the variation in rate of change. Extreme values on either end of the histogram indicate very rapid rises or drops in glucose—a high degree of local variability. In Fig 4, Subject 1 once again shows lower glucose variability by having a narrower histogram with most values falling between -2 mg/dl/min and 2 mg/dl/min. Subject 5 has a shorter, more widely distributed histogram indicating greater glucose variability. The function call to produce this plot is as follows.

```
hist_roc(example_data_5_subject, subjects = c("Subject 1",
"Subject 5"))
```

Finally, `iglu` allows one to generate an Ambulatory Glucose Profile (AGP) report in accordance with recommendations in [29]. Fig 5 shows an example report for Subject 1, which includes information on data collection period, time spent in standardized glycemic ranges (cutoffs of 54, 70, 180 and 250 mg/dL) displayed as a stacked bar chart [28], glucose variability as measured by %CV, and visualization of quantiles of the glucose profile across days together with daily glucose views.

## Relationship between metrics

To illustrate the relationships between different metrics and their interpretation, we calculated all metrics for example data of 5 subjects. Fig 6 shows the heatmap of resulting metrics

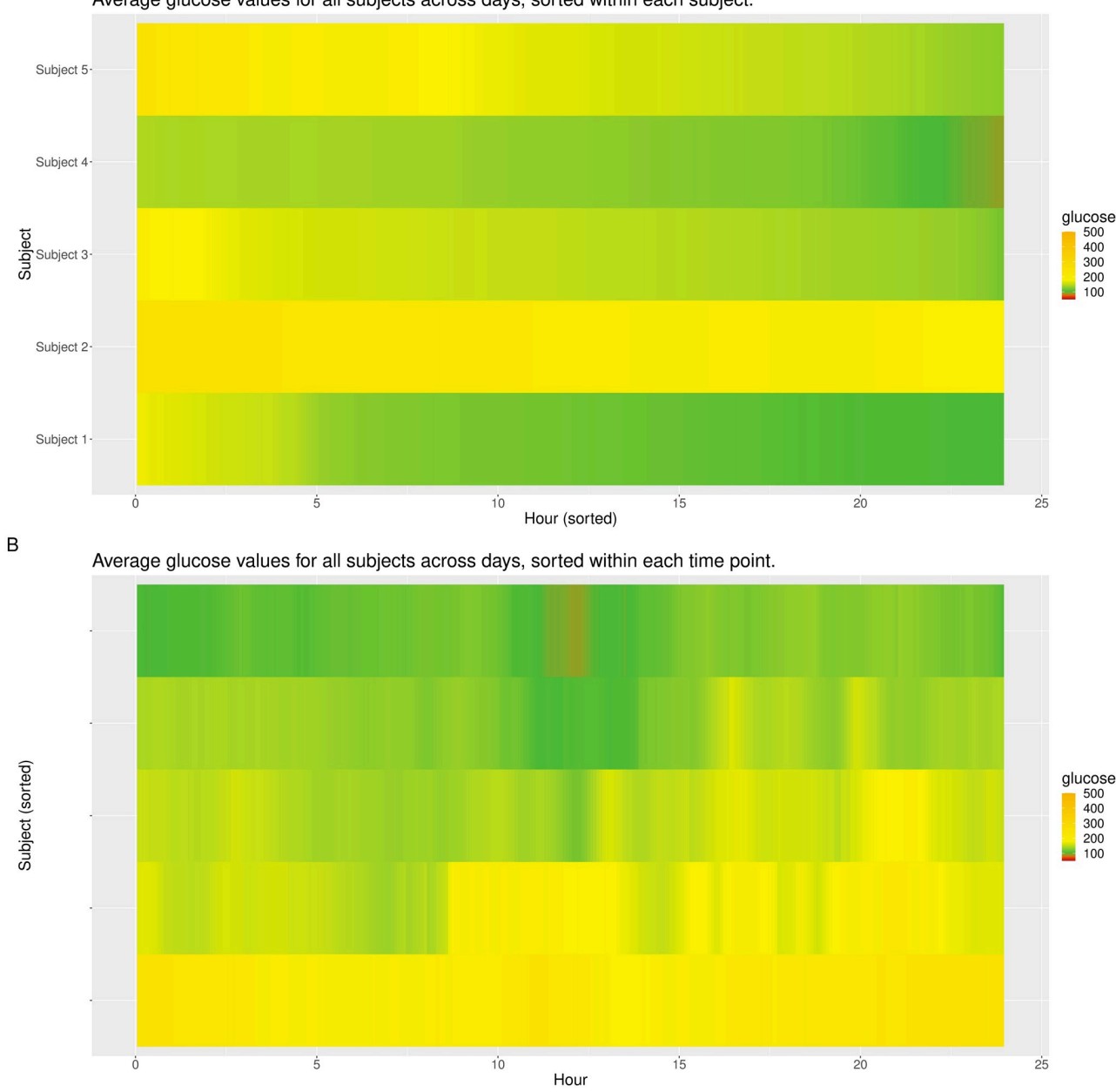

**Fig 3. Multi-subject lasagna plots in 'red-orange' color scheme. (A)** sorted within each subject and **(B)** sorted within each time point across subjects.

(centered and scaled across all subjects to aid visualization) created using R package pheat-map [30]. The hierarchical clustering of glucose metrics results in six meaningful groups with the following interpretation (from top to bottom): (1) in range metrics; (2) hypoglycemia metrics; (3) hyperglycemia metrics; (4) a mixture of variability and hyperglycemia metrics; (5) CVsd (standard deviation of CV, coefficient of variation, across days); (6) glucose variability metrics. Interestingly, while CVsd is a measure of glucose variability, it behaves quite differently from other variability metrics in these 5 subjects. The hierarchical clustering of subjects confirms our previous observations that Subject 2 has the worst hyperglycemia (highest values

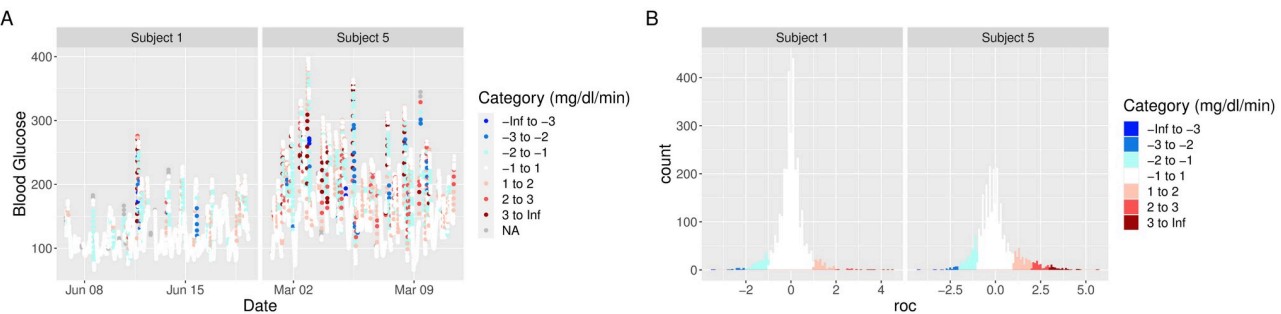

**Fig 4. Rate of change visualizations. (A)** time-series and **(B)** histogram plots of rate of change for two selected subjects from example dataset.

for metrics in group (2)), whereas Subject 5 has the highest glucose variability (highest values for metrics in group (6)). The relationship between a reduced list of metrics has also been studied in [31, 32] using sparse principal component analysis. While [31, 32] focus on selection of a few key metrics to describe glucose variability, our goal here is exploratory analysis to illustrate differences and similarities between all metrics on a given dataset.

## GUI via shiny application

The iglu package comes with a shiny application [33], which provides a point-and-click graphical user interface (GUI) for all metric calculations and visualizations. The interface can be accessed from R console by calling

```
iglu::iglu_shiny()
```

or directly at https://irinagain.shinyapps.io/shiny_iglu/. The users can load their CGM data in .csv format, and export metrics output to the user's clipboard or to .csv, .xlsx, or .pdf files (Fig 7A and 7B). Fig 7C shows an example of shiny interface for creating customized visualization plots based on user-loaded data.

## Conclusion

The iglu package is designed to simplify computations of CGM-derived glucose metrics, and assist in CGM data visualization. The current version includes all of the metrics summarized in [2] as well as many others (see Table 1). New metrics will be incorporated into the future versions as they develop. More details on the package functionality together with the full documentation are provided in the package website at https://irinagain.github.io/iglu/.

Several limitations exist when compared to existing CGM software. First, the R interface assumes that the CGM data is already loaded into R as a data frame, which requires users to have sufficient R knowledge for data processing. The Shiny app currently only allows one to load CGM data in .csv format, and thus it also requires initial pre-processing by the user, albeit not necessarily in R. This is not the case for CGManalyzer or cgmanalysis, which can work directly with specialized data formats from many popular CGMs. Nevertheless, continuous development of new CGM meters coupled with varying data formats across meters present definite challenges for any CGM software. Secondly, while the list of metrics implemented in iglu is more comprehensive compared to other R packages on CGM (Table 3), it still lacks some functionality that may be desired as part of the AGP output [6], specifically the count of hypoglycemia/hyperglycemia excursions, and separation of metrics into sleep/wake time periods. Thirdly, while the agreement of metric values across software packages is encouraging, it does not necessarily signify the agreement with gold standard (see also the discussion in [4]). Furthermore, a comprehensive cross-comparison across packages is quite difficult as it

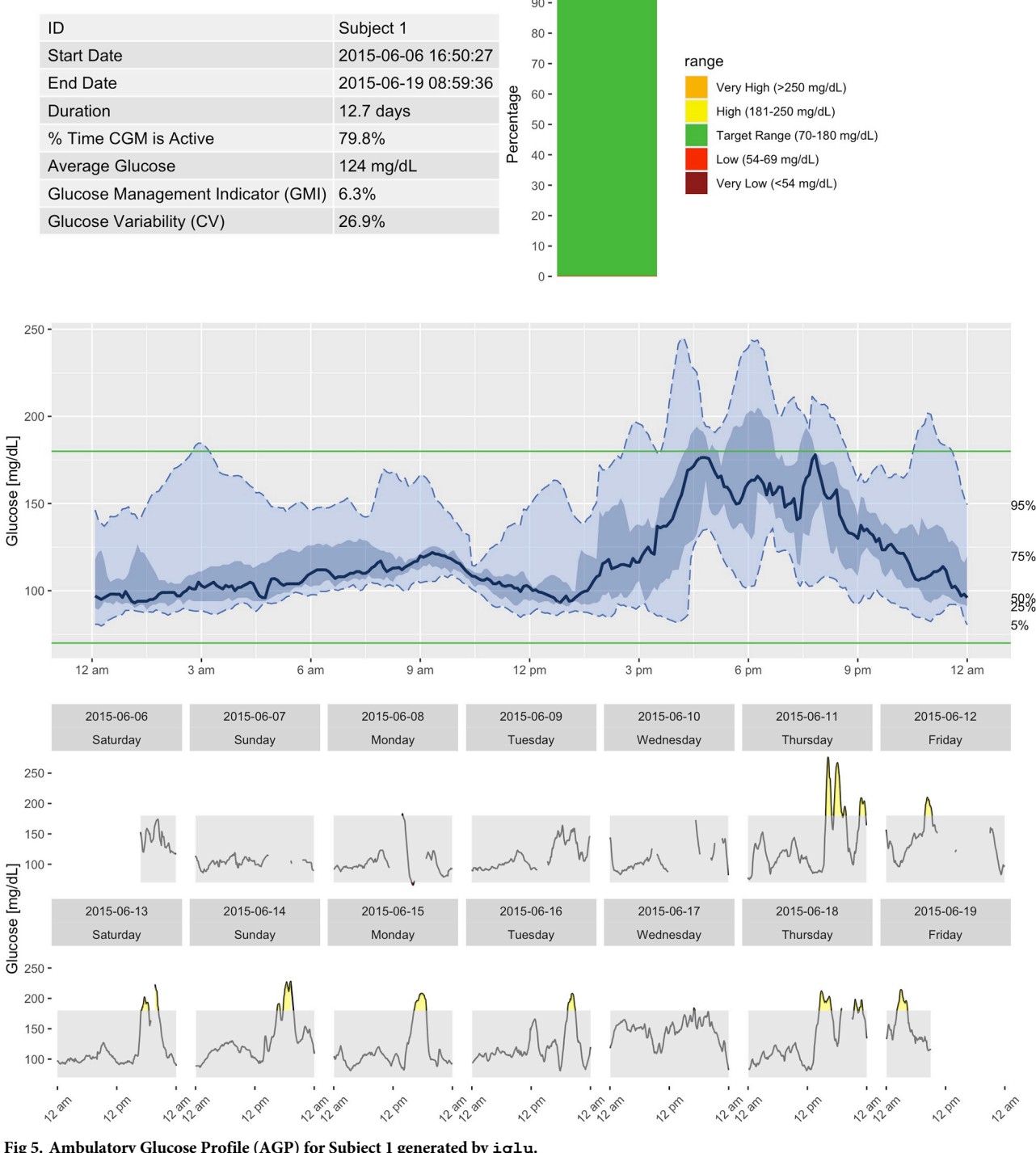

**Fig 5. Ambulatory Glucose Profile (AGP) for Subject 1 generated by `iglu`.**

requires a careful adjustment for potential differences in default parameters used in metrics calculations, in handling of missing values, and in underlying algorithms used. However, we believe that the explicit metric values provided in Table 3 coupled with public availability of our example dataset will serve as a useful preliminary step towards this endeavor. We hope to

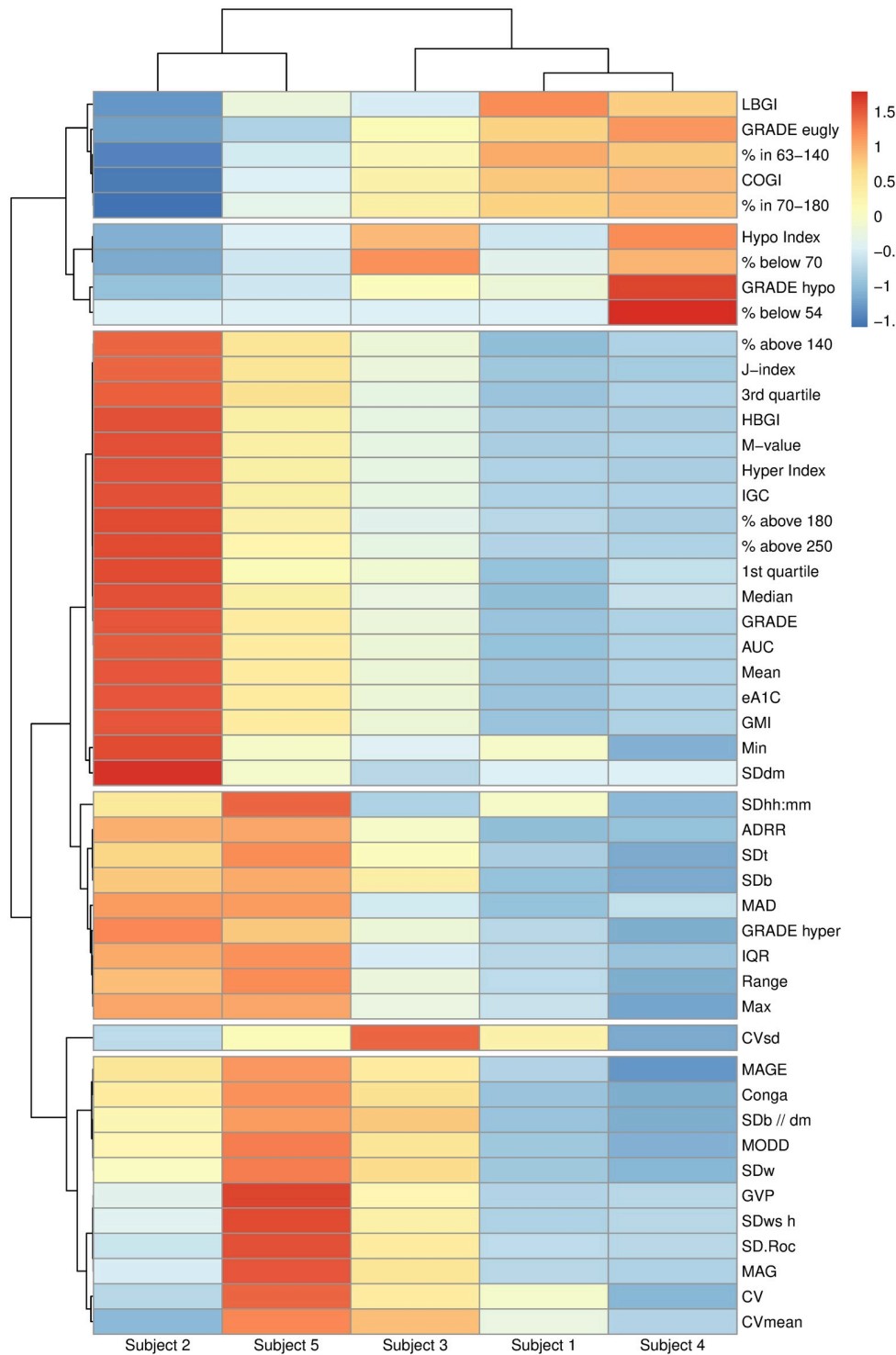

**Fig 6. Heatmap of all metrics calculated using `iglu` for 5 subjects with Type II diabetes.** Hierarchical clustering is performed on centered and scaled metric values using distance correlation and complete linkage. The cluster tree for metrics is cut at 6 groups, which can be interpreted as follows (from top to bottom): (1) in range metrics; (2) hypoglycemia metrics; (3) hyperglycemia metrics; (4) a mixture of variability and hyperglycemia metrics; (5) CVsd (standard deviation of CV, coefficient of variation, across days); (6) glucose variability metrics. The heatmap supports that Subject 2 has the worst hyperglycemia and Subject 5 has the highest glucose variability.

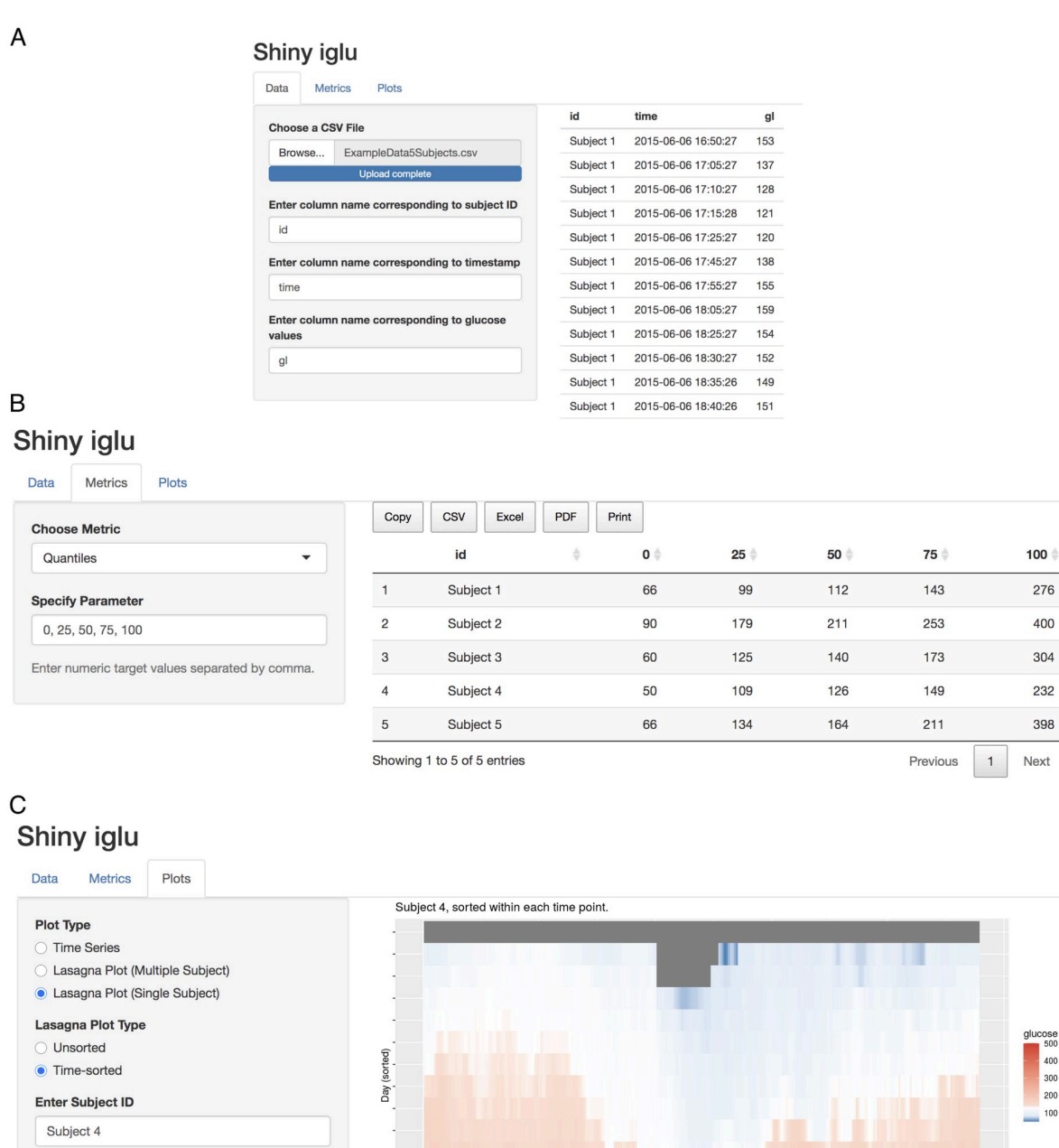

**Fig 7. Shiny GUI interface for `iglu`.** **(A)** loading CGM data in .csv format; **(B)** calculating user-specified quantiles for each subject; **(C)** creating customized lasagna plot for the selected subject.

address some of these limitations in future iterations by leveraging complimentary existing open-source CGM software and our own updates to `iglu`.

From the context of CGM data applications, we acknowledge that an extensive list of metrics may have little utility for individual patient care in day to day clinical practice beyond the commonly used % of glucose values in range, Mean, CV, etc. Despite this, the `iglu` has important strengths that merit discussion. First, given that CGM is likely to be incorporated in clinical studies outside the realm of Type 1 diabetes, having accessibility to methods for summarizing CGM measures that extend the typical panel is of value. Second, CGM measures provide dynamic characterization of glucose trajectories which can be of immense value when considering the potential impact of conditions that are associated with acute temporal changes in pathophysiological mechanisms that can impact glucose homeostasis. For example, sleep apnea is a common condition that affects 9% of women and 25% of men in the general population. It is well known that sleep apnea is associated with nocturnal repetitive increases in sympathetic activity due to cyclical hypoxemia and recurrent arousals from sleep. Thus, to determine whether these acute changes which are known to increase sympathetic nervous system activity can influence glucose homeostasis measures that capture the dynamic nature of glucose trajectories are needed. Even in clinical scenarios where acute pathophysiological changes are not present, metrics that help probe the temporal nature of glucose are of value. For example, obesity is associated with metabolic flexibility. Having detailed CGM measures that help define the various rate of change (increase and decrease in glucose levels) can provide insight into how conditions such as obesity and polycystic ovary syndrome, alter the diurnal nature of glucose profiles. Furthermore, given the detailed nature of CGM data and the increasing use of acquiring such data, we believe that convenient methods for analyzing CGM data are desperately needed to facilitate the use of CGM methodology by investigators in observational studies and randomized clinical trials.

In summary, while there are existing open-source R packages for CGM data analyses [3, 4], these packages focus more on CGM data reading than exhaustive metric implementation, and require programming experience. Instead, `iglu` focuses on comprehensive implementation of available CGM metrics and ease of use via accompanying GUI application. All data loading, parameter selection, metric calculations and visualizations are available via point-and-click graphical user interface. This makes `iglu` accessible to a wide range of users, which coupled with the free and open-source nature of `iglu` will help advance CGM research and CGM data analyses.

## Acknowledgments

The authors are thankful to Marielle Hicban, Mary Martin, Nhan Nguyen, Pratik Patel and John Schwenck for assisting in writing several metric functions in version 2.0.0 of `iglu` package.

## Author Contributions

**Conceptualization:** Steven Broll, John Muschelli, Irina Gaynanova.

**Data curation:** Naresh M. Punjabi.

**Formal analysis:** Steven Broll, Elizabeth Chun, Irina Gaynanova.

**Funding acquisition:** Naresh M. Punjabi, Irina Gaynanova.

**Investigation:** Irina Gaynanova.

**Methodology:** Irina Gaynanova.

**Project administration:** Irina Gaynanova.

**Resources:** Irina Gaynanova.

**Software:** Steven Broll, Jacek Urbanek, David Buchanan, Elizabeth Chun, John Muschelli, Irina Gaynanova.

**Supervision:** Irina Gaynanova.

**Validation:** David Buchanan, John Muschelli.

**Visualization:** Steven Broll, Elizabeth Chun, Irina Gaynanova.

**Writing – original draft:** Steven Broll, Irina Gaynanova.

**Writing – review & editing:** Jacek Urbanek, David Buchanan, Elizabeth Chun, John Muschelli, Naresh M. Punjabi, Irina Gaynanova.

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
