## [Decision Letter · Decision Letter 0]

20 Nov 2020

PONE-D-20-30578

Interpreting blood GLUcose data with R package iglu

PLOS ONE

Dear Dr. Gaynanova,

Thank you for submitting your manuscript to PLOS ONE. After careful consideration, we feel that it has merit but does not fully meet PLOS ONE’s publication criteria as it currently stands. Therefore, we invite you to submit a revised version of the manuscript that addresses the points raised during the review process.  Please note that Reviewer 2 made a number of suggestions for modification of the software which should be viewed as optional recommendations.  

I have a few comments to add to those of the reviewers:

1) Please perform a careful copy edit as there are typos and missing references.

2) In fig 2c, if these are 24 hour averages how can the x-axis unit be hours?

3) The Shiny interface is an extremely useful step, reducing the need for the user to have programming experience.  However, the user still needs to get the data into 3 column format, which is not the format exported by the CGM software.  Therefore, this step may require programming or manual editing of the data, which could introduce error.

We look forward to receiving your revised manuscript.

Kind regards,

Laura Pyle

Academic Editor

PLOS ONE

Journal Requirements:

2. In your ethics statement in the manuscript and in the online submission form, please provide additional information about the dataset used in your study. Specifically, please ensure that you have discussed whether all data were fully anonymized before you accessed them.

3.We suggest you thoroughly copyedit your manuscript for language usage, spelling, and grammar. If you do not know anyone who can help you do this, you may wish to consider employing a professional scientific editing service.  

Reviewers' comments:

Reviewer's Responses to Questions

**Comments to the Author**

1. Is the manuscript technically sound, and do the data support the conclusions?

Reviewer #1: Yes

Reviewer #2: Yes

2. Has the statistical analysis been performed appropriately and rigorously? 

Reviewer #1: No

Reviewer #2: Yes

3. Have the authors made all data underlying the findings in their manuscript fully available?

Reviewer #1: Yes

Reviewer #2: Yes

4. Is the manuscript presented in an intelligible fashion and written in standard English?

Reviewer #1: Yes

Reviewer #2: Yes

5. Review Comments to the Author

Reviewer #1: The authors present the R package “iglu” and accompanying Shiny app for visualization and analysis of continuous glucose monitor (CGM) data. The software calculates CGM metrics and generates figures not available in other packages, and is accessible to users with little programming experience. It is an excellent piece of software that will significantly ease the burden of analysis for many researchers.

Minor revisions for the paper:

1) There are a couple of small typos, so the paper could do with one more round of copy editing

2) In the interest of fairness, it would be worth including in Table 2 metrics calculated by CGManalyzer and cgmanalysis that are not included in iglu.

3) Please add a little bit more detail on how the package handles missing values. Does the CGMS2DayByDay() function automatically fill in missing data using linear interpolation? Or does it fill in with NA values? Also, the caption of Figure 1 claims that metric calculations are not affected by missing glucose values, but this cannot be true for all metrics so clarification is needed.

4) Figure 4 and the section on “relationship between metrics,” while interesting, seem beyond the scope of this manuscript. The hierarchical-clustering tool does not appear to be available in the Shiny app (although adding it could be extremely useful), which makes the inclusion of this clustered heatmap slightly confusing.

5) Please add some brief comparisons to demonstrate how closely this software agrees with other packages. This does not need to be an in-depth statistical analysis, as that would also be beyond the scope of the paper. However, the primary concern for many readers will be the accuracy of the calculations, so brief side-by-side comparisons between packages would be reassuring.

6) Please add some limitations to the conclusion section.

Minor suggestions for the software:

1) Although the graphing artifact (Figure 1, subject 2) is obvious in this example data, smaller artifacts may be less obvious in real-life data. Missing values should be blank space in the plots to avoid confusion.

2) After loading the 5 subject example data in R, writing to .csv, and loading into the Shiny app, the active_percent() function produces Error: unused arguments (180, 200, 250).

3) The visualizations are generally excellent as they are, but one suggestion would be to look into making the plots interactive using the plotly graphing library in R. This is not necessary for publication, but perhaps a recommendation for future updates.

4) Many researchers use packages like this to generate metrics for further analysis, so it might be nice to have the option to download all the metrics at once rather than manually combining results from multiple functions. Again, this is not a necessary change for publication, but simply a recommendation for the future.

Reviewer #2: Comments to the authors:

1. The paper is well written. This work goes beyond previous methods/software for analysis of CGM data. There is a need for such open source software. This software includes several metrics that were not included in previous software. The work is done well technically.

2. The introduction of Lasagna plots is new and novel. That approach is closely related to the use of stacked bar charts as introduced by Rodbard in 2009, and now included in nearly all outputs for CGM data.It remains to be see to what extent the Lasagna plot is superior to use of the stacked bar chart.

3. The authors note that they might add additional metrics in the future.

New metrics continue to be developed. See for instance Leelarathna et al, re COGI, published in 2020 in Diabetes Technology and Therapeutics.

4. The authors do not include an AGP in their outputs. This is perhaps the most popular component of nearly all displays of CGM data at the present time – along with the stacked bar chart. See Mazze et al, Diabetes Care 1987 and also 2008 2009 and multiple other papers, and subsequent papers from Bergenstal et al. See a recent review from Rodbard in press and online at DTT.

5. Most endocrinologists and other diabetes specialists use 70 and 180 as the cutoffs for in range. The present authors use 70 and 140. The graphs will look more familiar and appropriate to many readers if you use 70 and 180, or even 54 70 180 and 250. See Battelino et al, 2019, Diabetes Care. (63 and 140 are used or proposed for pregnancy).

6. It is reasonable for the authors to use their own nomenclature for their own variables in their program. However, for clarity, the authors might use the nomenclature as established in the literature for a number of variables, e.g. the subclassification of SD, i.e. SDT SDw SDb SDhh:mm SDdm SDb//dm and some others. e.g., bottom of page 4/9

7. Several of the metrics have ‘parameters’, i.e. TIR, TBR TAR MR CONGAn IGC (with a,b,c,d, LLTR ULTR). Please specify what default parameters you are using, and whether or not the end user can adjust those parameters. Cf top of page 4/9. Also SDwsh – what are the parameters for h? and is this for the entire series with all possible starting points, or for particular starting points?

8. Throughout - this reviewer prefers use of ‘allows one to map’ rather than ‘allow to map’ (several other instances for the word allows)

9. Page 5/9: under visualizations, line 5: reference to Section ___ (the number of the section is not currently specified)

10. The color scheme used by the authors is perfectly acceptable – but most of the commercial software for CGM data analysis uses a different color scheme. Authors might to well to build in flexibility and allow the end user to select their own color scheme.

11. Consider the possible option to include a logarithmic scale for glucose (Rodbard 2009 .. separate article than the ones cited).

12. Page 6/9: Authors discuss “worse glucose control” at at least two locations in the paper. But worse has several dimensions – can be in terms of TIR TBR TAR SD CV etc. So authors should be careful in using the term “worse” – it is “worse” in only 1 dimension.

13. Authors seem unaware of two articles by Fabris C et al (with Breton or Cobelli or Kovatchev) using principal components analysis and the high degree of correlation of many of the metrics. These probably should be cited, vis a vis the hierarchical analysis used by the present authors.

14. What algorithm are the authors using for MAGE (please use MAGE in all caps – all uppercase). Peter Baghurst has an algorithm (DTT) written in R. Are you using that one? That has been ‘validated’ or at least tested – see Sechterberger et al.

15. CVsd is an entirely new metric—what does it mean> What is it correlated with? Any place where the authors believe it might be helpful?

16. In your function CGMS2DaybyDay, what is the nature of your smoothing function? Any particular reason why it was selected or what its advantages might be relative to some alternatives? Any indication that it performs better than others.

17. You do not seem to include MAG (DeVries, Hermanides ?) also also see distance traveled (DT from Marling). A recent study from Moscada (?) and Nick Oliver indicates that MAG has the highest value for Discriminant Ratio, a criterion they propose as a basis to select among alternative metrics for various aspects of CGM.

18. Various consensus papers (Bergenstal, Danne, Maahs, Battelino) have discussed the statistics that the consensus groups believed should be presented together with the AGP (and other analyses of the CGM data). Please be sure that you have computed and provide the ones that are currently or recently in vogue.

19. Several authors have investigated how much data must be available to obtain reliable estimates of the parameters. These include Xing et al, Riddlesworth et al, and Nick Oliver or others from his group (may be in press or online at the present time, in DTT).

20. There are two major sets of applications for CGM data: The first is for the care of the individual patient. For this, one does not need or want all possible parameters… there is now a consensus that TIR TAR TBR Mean or median, %CV, and perhaps a very few other parameters are sufficient – in any event they are usually more than either the physician, health care practitioner, patient or patient’s family can digest. So, there is very little demand for new metrics – most unfamiliar, and most highly redundant or at least highly correlated with the few mentioned here.

21. The other major application of CGM data is in the context of clinical trials. Here one is interested in significance testing for superiority or non-inferiority for one or the other treatment, using randomized parallel studies or crossover studies. These often involve multivariate corrections e.g. for baseline A1C, or baseline mean glucose, and for clinical sites, and other subsets of the data. This is what a pharmaceutical laboratory would need. The present article, and program, despite all of its strengths, does not really fulfil the needs to these two major potential groups of users (in my opinion). The current study and paper is still valuable, and publishable, but this reviewer would suggest that the authors try to address or at least acknowledge these two major use-cases, and either try to address them in the present paper, or address them in future studies and manuscripts. Perhaps the lack of the ability of the present study and code to address these important use-cases should be included as a limitation of the study. However, the present code should be useful to others trying to address those cases.

22. Hill and Oliver et al have recently published (or have online) a new version of their EZ GV method (still in Excel, I believe) that might be helpful.

Some of the references cited in my comments above: Some of these might be helpful to the authors in the context of the present paper; others may be helpful in the future, to enable them to better coordinate with the “clinical diabetes” side of the relevant literature.

1. Updated Software for Automated Assessment of Glucose Variability and Quality of Glycemic Control in Diabetes.

Moscardó V, Giménez M, Oliver N, Hill NR.Diabetes Technol Ther. 2020 Oct;22(10):701-708. doi: 10.1089/dia.2019.0416. Epub 2020 Apr 22.PMID: 32195607

2. Assessment of Glucose Control Metrics by Discriminant Ratio.

Moscardó V, Herrero P, Reddy M, Hill NR, Georgiou P, Oliver N.Diabetes Technol Ther. 2020 Oct;22(10):719-726. doi: 10.1089/dia.2019.0415.PMID: 32163723

3. Calculating the mean amplitude of glycemic excursion from continuous glucose monitoring data: an automated algorithm.

Baghurst PA.Diabetes Technol Ther. 2011 Mar;13(3):296-302. doi: 10.1089/dia.2010.0090. Epub 2011 Feb 3.PMID: 21291334

4. Ambulatory glucose profile: representation of verified self-monitored blood glucose data

R S Mazze, D Lucido, O Langer, K Hartmann, D Rodbard PMID: 3552508 DOI: 10.2337/diacare.10.1.111

5. Recommendations for standardizing glucose reporting and analysis to optimize clinical decision making in diabetes: the Ambulatory Glucose Profile (AGP).

Bergenstal RM, Ahmann AJ, Bailey T, Beck RW, Bissen J, Buckingham B, Deeb L, Dolin RH, Garg SK, Goland R, Hirsch IB, Klonoff DC, Kruger DF, Matfin G, Mazze RS, Olson BA, Parkin C, Peters A, Powers MA, Rodriguez H, Southerland P, Strock ES, Tamborlane W, Wesley DM.Diabetes Technol Ther. 2013 Mar;15(3):198-211. doi: 10.1089/dia.2013.0051. Epub 2013 Feb 28.PMID: 23448694

6. J Diabetes Sci Technol . 2009 Nov 1;3(6):1395-401.

doi: 10.1177/193229680900300620. A semilogarithmic scale for glucose provides a balanced view of hyperglycemia and hypoglycemia David Rodbard 1 PMID: 20144394 PMCID: PMC2787040 DOI: 10.1177/193229680900300620

7. Clinical Targets for Continuous Glucose Monitoring Data Interpretation: Recommendations From the International Consensus on Time in Range.

Battelino T, Danne T, Bergenstal RM, Amiel SA, Beck R, Biester T, Bosi E, Buckingham BA, Cefalu WT, Close KL, Cobelli C, Dassau E, DeVries JH, Donaghue KC, Dovc K, Doyle FJ 3rd, Garg S, Grunberger G, Heller S, Heinemann L, Hirsch IB, Hovorka R, Jia W, Kordonouri O, Kovatchev B, Kowalski A, Laffel L, Levine B, Mayorov A, Mathieu C, Murphy HR, Nimri R, Nørgaard K, Parkin CG, Renard E, Rodbard D, Saboo B, Schatz D, Stoner K, Urakami T, Weinzimer SA, Phillip M.Diabetes Care. 2019 Aug;42(8):1593-1603. doi: 10.2337/dci19-0028. Epub 2019 Jun 8.PMID: 31177185 Free

8. International Consensus on Use of Continuous Glucose Monitoring. Danne T, Nimri R, Battelino T, Bergenstal RM, Close KL, DeVries JH, Garg S, Heinemann L, Hirsch I, Amiel SA, Beck R, Bosi E, Buckingham B, Cobelli C, Dassau E, Doyle FJ 3rd, Heller S, Hovorka R, Jia W, Jones T, Kordonouri O, Kovatchev B, Kowalski A, Laffel L, Maahs D, Murphy HR, Nørgaard K, Parkin CG, Renard E, Saboo B, Scharf M, Tamborlane WV, Weinzimer SA, Phillip M.Diabetes Care. 2017 Dec;40(12):1631-1640. doi: 10.2337/dc17-1600.PMID: 29162583 Free PMC article. Review

9. Outcome Measures for Artificial Pancreas Clinical Trials: A Consensus Report. Maahs DM, Buckingham BA, Castle JR, Cinar A, Damiano ER, Dassau E, DeVries JH, Doyle FJ 3rd, Griffen SC, Haidar A, Heinemann L, Hovorka R, Jones TW, Kollman C, Kovatchev B, Levy BL, Nimri R, O'Neal DN, Philip M, Renard E, Russell SJ, Weinzimer SA, Zisser H, Lum JW.Diabetes Care. 2016 Jul;39(7):1175-9. doi: 10.2337/dc15-2716.PMID: 27330126 Free PMC article. Review

10. Evaluating Glucose Control With a Novel Composite Continuous Glucose Monitoring Index. Leelarathna L, Thabit H, Wilinska ME, Bally L, Mader JK, Pieber TR, Benesch C, Arnolds S, Johnson T, Heinemann L, Hermanns N, Evans ML, Hovorka R.J Diabetes Sci Technol. 2020 Mar;14(2):277-283. doi: 10.1177/1932296819838525. Epub 2019 Mar 31.PMID: 30931606 Free PMC article.

11. A Review of Continuous Glucose Monitoring-Based Composite Metrics for Glycemic Control. Nguyen M, Han J, Spanakis EK, Kovatchev BP, Klonoff DC.Diabetes Technol Ther. 2020 Aug;22(8):613-622. doi: 10.1089/dia.2019.0434. Epub 2020 Mar 4.PMID: 32069094

6. PLOS authors have the option to publish the peer review history of their article (what does this mean?). If published, this will include your full peer review and any attached files.

Reviewer #1: **Yes: **Tim Vigers

Reviewer #2: No

---

## [Author Response · Author response to Decision Letter 0]

1 Feb 2021

The detailed response to reviews is attached as pdf with this submission

---

## [Decision Letter · Decision Letter 1]

2 Mar 2021

Interpreting blood GLUcose data with R package iglu

PONE-D-20-30578R1

Dear Dr. Gaynanova,

We’re pleased to inform you that your manuscript has been judged scientifically suitable for publication and will be formally accepted for publication once it meets all outstanding technical requirements.

Kind regards,

Laura Pyle

Academic Editor

PLOS ONE

Additional Editor Comments (optional):

The second reviewer notified the Editor by email that all comments had been addressed.

Reviewers' comments:

Reviewer's Responses to Questions

**Comments to the Author**

1. If the authors have adequately addressed your comments raised in a previous round of review and you feel that this manuscript is now acceptable for publication, you may indicate that here to bypass the “Comments to the Author” section, enter your conflict of interest statement in the “Confidential to Editor” section, and submit your "Accept" recommendation.

Reviewer #1: All comments have been addressed

2. Is the manuscript technically sound, and do the data support the conclusions?

Reviewer #1: Yes

3. Has the statistical analysis been performed appropriately and rigorously? 

Reviewer #1: Yes

4. Have the authors made all data underlying the findings in their manuscript fully available?

Reviewer #1: Yes

5. Is the manuscript presented in an intelligible fashion and written in standard English?

Reviewer #1: Yes

6. Review Comments to the Author

Reviewer #1: (No Response)

7. PLOS authors have the option to publish the peer review history of their article (what does this mean?). If published, this will include your full peer review and any attached files.

Reviewer #1: **Yes: **Tim Vigers

---

## [Editor Report · Acceptance letter]

5 Mar 2021

PONE-D-20-30578R1 

Interpreting blood GLUcose data with R package iglu  

Dear Dr. Gaynanova:

I'm pleased to inform you that your manuscript has been deemed suitable for publication in PLOS ONE. Congratulations! Your manuscript is now with our production department. 

Kind regards, 

on behalf of

Dr. Laura Pyle 

Academic Editor

PLOS ONE